# Individual Treatment Trials—Do Experts Know and Use This Option to Improve the Treatability of Mucopolysaccharidosis?

**DOI:** 10.3390/ph16030416

**Published:** 2023-03-09

**Authors:** Anna-Maria Wiesinger, Hannah Strobl, Florian B. Lagler

**Affiliations:** 1Department of Medical Science, Institute of Congenital Metabolic Diseases, Paracelsus Medical University, 5020 Salzburg, Austria; 2Department of Rare Diseases, European Reference Network for Hereditary Metabolic Diseases, MetabERN, 33100 Udine, Italy

**Keywords:** mucopolysaccharidosis, MPS, individual treatment trials, n-of-1, decision analysis framework model, expert survey, ESITT, rare disease treatment, precision medicine

## Abstract

Mucopolysaccharidoses (MPS) are a group of rare, heterogeneous, lysosomal storage disorders. Patients show a broad spectrum of clinical features with a substantial unmet medical need. Individual treatment trials (ITTs) might be a valid, time- and cost-efficient way to facilitate personalized medicine in the sense of drug repurposing in MPS. However, this treatment option has so far hardly been used—at least hardly been reported or published. Therefore, we aimed to investigate the awareness and utilization of ITTs among MPS clinicians, as well as the potential challenges and innovative approaches to overcome key hurdles, by using an international expert survey on ITTs, namely, ESITT. Although 74% (20/27) were familiar with the concept of ITTs, only 37% (10/27) ever used it, and subsequently only 15% (2/16) published their results. The indicated hurdles of ITTs in MPS were mainly the lack of time and know-how. An evidence-based tool, which provides resources and expertise needed for high-quality ITTs, was highly appreciated by the vast majority (89%; 23/26). The ESITT highlights a serious deficiency of ITT implementation in MPS—a promising option to improve its treatability. Furthermore, we discuss the challenges and innovative approaches to overcome key barriers to ITTs in MPS.

## 1. Introduction

Mucopolysaccharidoses (MPS) are a group of congenital metabolic storage diseases, with a highly location-specific prevalence rate of 1.53 per 100,000 live births [1]. Due to an enzyme deficiency, MPS are leading to an accumulation of glycosaminoglycans (GAGs) in lysosomes and the extracellular matrix (ECM) [2,3,4]. Depending on the missing or transforming enzyme, MPS are classified into seven main types and multiple subtypes with a different and broad range of somatic, skeletal, and central nervous system (CNS) clinical features [4,5,6].

To date, there are no satisfactory or curative therapies for any of the known types of MPS available, despite the enormous unmet medical need among patients. The two current approved standard therapies are enzyme replacement therapy (ERT), for MPS I, II, IVA, VI, and VII, and hematopoietic stem cell transplantation (HSCT), mainly for MPS I [6,7]. The injected enzymes of ERT cannot cross the blood–brain barrier (BBB); thus, this treatment option has no success in the CNS and solely represents a general, causal treatment attempt for somatic clinical features [8,9,10]. HSCT improves the quality of life with the ability to alleviate the course of the disease. It is more beneficial in patients younger than 16 months of age because irreversible damage to bone that occurred before transplantation remains [11]. The long procedure, the effort to find a suitable donor, and multiple adverse effects, such as infections, organ failure, or graft rejection, comprise serious difficulties and challenges of HSCT [12,13].

Given the wide spectrum of disease severity, the varying combinations of progressive multisystem clinical features, and the lack of potential therapies, with ERT and HSCT as the only approved options, MPS patients ultimately face a substantial unmet medical need, associated with reduced life expectancy and deteriorated quality of life.

To date, approximately 7000 rare diseases are globally known [14], all confronted with a considerable need for successful therapeutic options. Therefore, off-label drug use might be a promising alternative for MPS and other diseases. Off-label drugs are medications which are prescribed outside of their clinical approved authorization regarding dose, age, or indication [15,16]. Recently, Schrier et al. reported that off-label therapies in children are distributed differently in European countries [17]. The numbers confirm that 13–69% of pediatrics and 6–72% of adult hospitalized patients were prescribed an off-label therapy [18]. Similarly, Gore et al. stated that 9–78.7% of off-label drug use is by pediatric patients [19]. Despite the high unmet medical need in MPS, this option receives little attention. Likewise, off-label use successes or failures in rare diseases are rarely reported or published [14].

Individual treatment trials (ITTs), also known as n-of-1 trials, are the experimental use of a novel treatment method or drug, i.e., the use of a treatment without scientifically proven efficacy or outside indications of patients, where established treatment methods no longer help [20,21]. This provides the best opportunities for clinicians by enabling them to address the individual responses of each patient to find the optimal therapy [22,23]. Therefore, ITTs can be extremely useful for investigating remarkable findings and new effects of care [24] in chronic or rare diseases [25,26,27] with unsatisfactory therapies, in order to be able to increase the efficacy of the treatment [28].

Nevertheless, success is not achieved and guaranteed in all uses, because cumulative and curative treatments are unsuitable for ITTs. More progress can be expected with drugs that have a quick onset of action and a quick wash out [25,29]. These ITTs, which are often crossover trials, double-blinded, and randomized [26,28,30], can be conducted in three different ways. In multi-crossover trials, two treatments (A and B) are always tested differently. In the first type, only two phases (A–B, B–A) are performed, whereas in the second option, these two treatments are repeated several times in various combinations (A–A, B–A, A–B, B–B). Both times that the treatments are given one after the other, they are washed out and changed after a short period of time to see which treatment brings the best effect. The third option is called pre-post trial, where a treatment is performed on the patient and the condition before and after is compared [31].

Recently, it has been recognized that inflammatory processes play a major role in MPS. This offers a range of targets to intervene with immunomodulators, such as anakinra, adalimumab, or abatacept, as potential adjuvant therapeutic options [32]. However, research regarding adjuvant therapeutic options is particularly hampered in MPS. Conducting clinical trials in rare diseases is often fraught with risk and uncertainty due to the heterogeneity and the small number of patients [26,27,33]. A clinically highly relevant therapy response must be detected individually, especially in MPS. This makes ITTs an excellent alternative as they can easily overcome the burden for clinical trials, with the possibility to consider the respective risk and efficacy in each patient individually. Unfortunately, this possibility is hardly used or at least hardly reported in scientific peer-reviewed journals, despite the high unmet medical need in MPS.

## 2. Results

### 2.1. Sociodemographic Data

A total of 28 physicians from Europe and America completed the ESITT, and finally 27 were included in the analysis, with 20 participants (74.07%) from European countries, 6 (22.22%) from South America, and 1 (3.70%) from the USA.

The study population solely contained practicing physicians with profound knowledge in the field of MPS, with 10 males (37.04%) and 17 females (62.96%). Sociodemographic data are summarized in detail in Table 1 below and in the Appendix A.

### 2.2. Contentment with Available Therapy Options

Most participants expressed their dissatisfaction with the available therapies for MPS. Only 3 (11.11%) physicians responded that they are satisfied with the available therapies, whereas 44.44% (*n* = 12) disagreed and 25.93% (*n* = 7) strongly disagreed. Thus, roughly 70% of physicians are unsatisfied with the available therapies. The majority of the included clinicians (*n* = 15, 55.56%) also stated the dissatisfaction of their patients. Data and related questions are listed in Table 2.

### 2.3. Familiarity and Utilization of ITTs

Most participants (*n* = 20, 74.07%) surveyed had already heard of ITTs and were familiar with this topic. However, more than a quarter of the study population (*n* = 7, 25.93%) indicated a lack of confidence and experience regarding ITTs, as depicted in Figure 1.

Only 10 (37.04%) participants reported that they had already performed ITTs as a treatment strategy in MPS patients or implemented ITTs in patients with other diseases (*n* = 10, 37.04%). In total, only 14 clinicians had ever used ITTs in the treatment of their patients, as 6 study participants overlapped in the utilization of ITTs in MPS as well as in other conditions.

We further analyzed the types and subtypes of the enrolled MPS patients in ITTs. Participating clinicians (*n* = 8) reported that they had already conducted ITTs with MPS patients affected by MPS I Hurler (*n* = 2, 25.00%), MPS II (*n* = 3, 37.50%), MPS III B (*n* = 1, 12.50%), MPS VI (*n* = 1, 12.50%), and MPS VII (*n* = 1, 12.50%). Two physicians did not provide information.

Furthermore, we asked how many ITT clinicians (*n* = 9) had already implemented the treatment of MPS patients. The vast majority (*n* = 7, 77.78%) reported a single use. The remaining 22.22% of the study population performed two (*n* = 1) or three (*n* = 1) ITTs. One participant did not provide an answer.

Of a total of 14 physicians who indicated a previous utilization of ITTs in MPS or other diseases, only 4 out of 12 (33.33%) used a monitoring plan. Two physicians did not provide information. Moreover, the majority did not involve other experts in the ITT implementation (*n* = 5/13, 38.46%) and did not report or publish their ITT results in a peer-reviewed scientific journal (*n* = 2/13, 15.38%). One clinician did not provide information on the previous two questions. Three out of four participants who used a monitoring plan also worked together with experts. Participants who reported that they had published their ITT results (*n* = 2, 15.38%) had performed ITTs in patients with other diseases. Up to now, no results on ITTs in MPS have been reported successfully.

The two main reasons for not or hardly performing any ITTs in MPS were the impracticability of implementation (*n* = 5, 23.81%) and presumed irrelevance for the affected patients (*n* = 5, 23.81%). Other experts indicated an insufficiency in training (*n* = 4, 19.05%) and time considerations (*n* = 4, 19.05%), and 3 participants (14.29%) chose other reasons as an answer. The details are clearly summarized in Table 3 and Figure 2.

### 2.4. Willingness to Use a Decision Analysis Framework (DAF) Tool for ITTs in MPS

The final survey question was related to an evidence-based DAF tool. Participants were asked to consider a scenario in which a free service exists to make data-driven treatment choices available more easily and on a rational basis in order to offer ITTs to selected MPS patients. Subsequently, the enrolled clinicians were asked if they would use this novel DAF tool to carry out at least one ITT per year. The vast majority, 88.46% (*n* = 23), were either highly likely or likely to use a service like this. Two physicians (7.69%) indicated that they would rather not perform ITTs despite the availability of a DAF tool. The data are shown in Figure 3 and Table 4.

The applied Kruskal–Wallis test for the group comparison of the survey questions “Have you ever heard of n-of-1 trials?” and “How likely are you to use a service like this to make data driven treatment choices at least once in the next year?” revealed a *p*-value of 0.50 (H = 0.45, χ^2^_c_ = 3.84, df = 1), meaning that there was no difference between the groups.

Similarly, a Kruskal–Wallis test was applied for a comparison of the previous utilization of ITTs in MPS and willingness to use an evidence-based DAF tool. No significant association (H = 0.97, *p* = 0.32, χ^2^_c_ = 3.84, df = 1) between these groups (yes, no) and their responses regarding their willingness to use a DAF tool were detected. The data and results supported the null hypothesis.

## 3. Discussion

This is the first systematic analysis of ITTs in MPS—an option to improve the treatability of patients with this rare, heterogeneous group of disorders. The results of our survey highlight that the majority of MPS clinicians are unsatisfied with the approved treatment options. Affected patients and their families also share this view. An ITT as a treatment strategy is well known to these experts, however little use is made of it. We identified the lack of know-how and resources as the most important barriers. However, if the selection of eligible patients, the assessment of the best drug candidate, and an evidence-based benefit–risk assessment were facilitated, almost 90% of clinicians would offer ITTs to their MPS patients. These results indicate that with a service which facilitates evidence-based clinical decision making and overcomes the identified hurdles of ITTs, innovative treatments such as immunomodulation may be utilized more often, resulting in an improved treatability of MPS.

ITTs are an increasingly recommended option to improve treatability in different settings of high unmet medical needs, including oncology and osteoarthritis [34,35,36,37,38,39,40,41,42,43,44,45,46]. However, all publications report underutilization of this promising option [34,35,36,37,38,39,40,41,42,43,44,45,46]. To the best of our knowledge, our study is the first which systematically assesses the actual utilization of and barriers to ITTs in a specific indication. In accordance with previous publications, we found that 37% of our MPS experts had conducted ITTs—4 out of 27 clinicians used it in MPS patients, 6 in MPS and other diseases, and 4 in patients with other disorders.

As the main barriers to conducting ITTs, our survey identified the unfeasibility of ITT implementation as the key challenge, followed by a lack of time and expertise. This is in line with the assumptions of other authors that proposed a lack of know-how and resources for the risk–benefit assessment. Furthermore, disruption of the patient–physician relationship, the additional effort for patients, and the low scientific validity of ITT results have been discussed [46,47,48,49,50]. The latter may explain our finding that ITTs were hardly reported (2/13; 15%) by our experts. As potential solutions to overcome these barriers, other authors proposed practical checklists, well-coordinated trainings, or standard protocol templates for physicians [47,48,50]. In accordance with that, the vast majority of our experts stated that they would offer ITTs if expertise and resources were facilitated, e.g., by a service or tool.

In general, expert surveys can be limited by several biases. Yet, we consider our study an important foundation for further research, as it is the first systematic assessment of the utilization of and barriers to ITTs in MPS. The representativeness of the participating experts is high given the 27 experts from 25 centres in 11 countries, managing 1100 patients overall, which is a considerable sample size (see Appendix A). Moreover, most questions have been answered quite homogenously. Consequently, despite the known limits of the applied methods, we find a strong indication for the development of a tool that facilitates evidence-based decision making in the frame of ITTs in MPS.

In conclusion, we can summarize that such support is likely to have a decisive impact on the utilization of ITTs in MPS, resulting in improved treatability. We are well aware that ITTs in the sense of off-label use can be hampered by the denial of reimbursement by payers or approval by ethical committees. Therefore, we plan to provide additional support with template documents such as assessment plans, informed consent, claim of reimbursement, etc. For smooth running and raising awareness, this project should be implemented with exclusive partnerships between academia and commercial and patient organizations with regulatory incentives.

Furthermore, the above-cited publications indicate that there is a substantial need for ITTs in many clinical fields and indications. Thus, our results may be applicable to and useful for similar projects in these fields.

## 4. Methods

### 4.1. Sampling and Eligibility Criteria

After a comprehensive literature search concerning ITTs in MPS, the questionnaire-based survey was implemented as a research instrument for gathering primary data. In the pilot phase, five top international MPS clinicians were selected for participation, in order to gain a first-hand insight and meaningful feedback.

In the second phase, the ESITT project was shared via personal networks, Metab-Latam, and MetabERN (Latin American and European Reference Network for Hereditary Metabolic Disorders), including an abstract and link for the online survey (software: SurveyMonkey^®^, San Mateo, CA, USA). Inclusion criteria for the study population were sufficient experience with MPS with a routine care of MPS patients and several years of clinical practice. Probands with no or less ITT experience had the possibility to skip several questions, which were specially marked. A total of 28 MPS experts completed the ESITT, and overall 27 MPS experts were enrolled in the evaluation. Furthermore, one questionnaire was excluded from the analysis due to inconsistencies and lack of experience in treating MPS patients.

### 4.2. Data Sources and Instruments

The developed cross-sectional survey was anonymous and voluntary and included predetermined answers and open questions. The time frame to complete the questionnaire was from December 2021 to June 2022. Overall, the ESITT can be divided into four different main sections: (i) personal characteristics; (ii) experience and satisfaction with currently approved MPS therapies; (iii) knowledge, awareness, utilization, and concerns regarding ITTs in MPS and other diseases; and (iv) necessity of a DAF tool for ITTs in MPS.

The questionnaire did not contain inversely worded items, which meant that no recoding was necessary. In order to enable counting, numbers were assigned to the variables by coding. The bipolar scale values were modified into unipolar scale values for evaluation. The response options of the items ran in a positive direction and were confirmed with high values. The questionnaire of the ESITT was adopted for MPS clinicians according to a validated tool [29] and is included as Appendix A.

#### 4.2.1. Sociodemographic Data

Sociodemographic data included variables that provide information about age, gender, and occupational situation of the participants.

#### 4.2.2. Satisfaction with Currently Approved MPS Therapies

Two questions of the ESITT focused on the perception of MPS clinicians regarding approved MPS treatment strategies, the first about their own satisfaction with the approved therapies and the second about the patient’s satisfaction. There were five possible answers on a bipolar scale ranging from strongly agree to strongly disagree.

#### 4.2.3. Knowledge, Awareness, Utilization, and Concerns Regarding ITTs

First, existing knowledge and familiarity with ITTs were assessed. The question “Have you ever heard of n-of-1 trials?” was evaluated by a dichotomous response format, ranging from yes to no. The next questions asked about experience and utilization of ITTs—generally and in MPS patients. Applied investigative medications in previous ITTs could be indicated by open questions. The main reasons why ITTs have been performed in practice could be selected by choosing one of six given answers. Furthermore, personal concerns regarding ITTs were evaluated by verifying whether previous ITTs had already been published, using dichotomous categories.

#### 4.2.4. Necessity of a DAF Tool for ITTs in MPS

The perception of and demand for a novel DAF tool for ITTs in MPS were surveyed with the question “Assume there is a service that made it easy for you to offer n-of-1 trials to select patients in your practice with MPS. How likely are you to use a service like this to make data driven treatment choices at least once in the next year?”. Respondents could indicate their willingness on a 5-point bipolar verbal scale from highly likely to highly unlikely.

### 4.3. Data Analysis and Synthesis

For the statistical analysis, the obtained survey responses were inserted into the Microsoft Excel program and the analytical software IBM^®^ SPSS^®^ Statistics (Chicago, IL, USA). Quantitative data were represented as numbers and percentages. A significance level of 5% for testing differences in variables was applied.

Descriptive statistics were performed for continuous variables by calculating minimal, maximal, mean, and standard deviation values. Using the same methods, the demand for a DAF tool for evidence-based, quantitative risk–benefit assessment in MPS was evaluated.

In hypothesis testing, the null hypothesis and an alternative hypothesis were put forward. The Kruskal–Wallis Test was performed as a non-parametric test, resulting in calculations of the *p*-value. The impact on willingness to use a DAF tool was analyzed by two different groups, with a negative attitude (no) and agreeing attitude (yes), by means of two questions—“Have you ever heard of n-of-1 trials?” and “Have you ever used an n-of-1 trial in the treatment of your MPS patients?”

### 4.4. Ethics Approval

ESITT was conducted in accordance with the relevant principles of the International Conference on Harmonization and Good Clinical Practice and was approved by the Ethical Committee in Salzburg, Austria (1125/2021; 5 July 2021).

## 5. Conclusions

This is the first systematic analysis of ITTs in MPS patients. Our study offers meaningful insights into clinicians’ awareness of, experience with, and willingness to conduct ITTs in MPS. Although the vast majority were familiar with the concept of ITTs, the experience was limited as well as the subsequent publication rate of realized ITTs. Our systematic analysis highlights that both clinicians and patients are unsatisfied with the approved therapies and that clinicians are highly willing to use ITTs more often if expertise and resources are provided with a free service. This study serves as an important starting point to address the substantial unmet medical need in MPS in a rational and personalized way.

## Figures and Tables

**Figure 1 pharmaceuticals-16-00416-f001:**
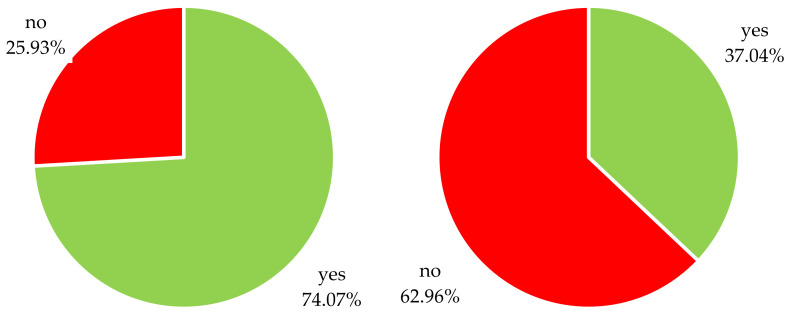
Response to the survey question “Have you ever heard of n-of-1-trials?” (left pie chart) and response to the survey question “Have you ever used n-of-1 trials in the treatment of your MPS patients?” (right pie chart); *n* = 27.

**Figure 2 pharmaceuticals-16-00416-f002:**
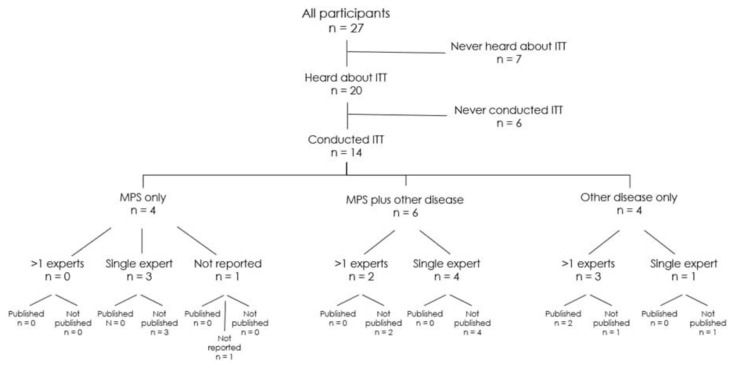
Flow chart overview of all participating MPS clinicians with regard to previous know-how and implementation of ITTs. Out of the 27 experts, 20 had already heard about ITTs and 14 had conducted ITTs—either with MPS patients or MPS and another disorder or only other disorders. A “single expert” is defined as a standalone expert, while “>1 expert” means a kind of interdisciplinary expert team.

**Figure 3 pharmaceuticals-16-00416-f003:**
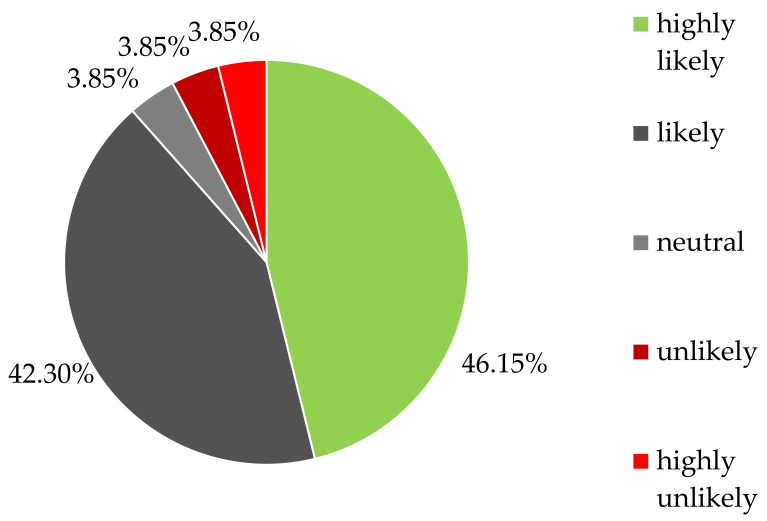
Response to the survey question “Assume there is a service that made it easy for you to offer n-of-1 trials to select patients in your practice with MPS. How likely are you to use a service like this to make data driven treatment choices at least once in the next year?” (*n* = 26).

**Table 1 pharmaceuticals-16-00416-t001:** Summary of the personal characteristics of MPS experts surveyed.

Age (*n* = 25)	
Mean age (standard deviation) Median age Minimum, Maximum	$\bar{x}\;$= 54.36 (10.04) $\tilde{x}$ = 53 Min = 36, Max = 70
Sex (*n* = 27)	
Female (*n*, %)	17 (62.96)
Male (*n*, %)	10 (37.04)
Years of clinical practice (*n* = 27)	
0–10 years (*n*, %)	2 (7.41)
11–20 years (*n*, %)	6 (22.21)
21–30 years (*n*, %)	11 (40.74)
31–40 years (*n*, %)	6 (22.22)
>40 years (*n*, %)	2 (7.41)
Number of patients treated (*n* = 27)	
<10 (*n*, %)	7 (25.93)
11–20 (*n*, %)	7 (25.93)
21–50 (*n*, %)	8 (29.63)
51–100 (*n*, %)	3 (11.11)
>100 (*n*, %)	2 (7.41)

**Table 2 pharmaceuticals-16-00416-t002:** Satisfaction of patients and physicians with current treatment options.

I Am Satisfied with the Available Treatment Options for My Patients with MPS. (*n* = 27)	
Strongly agree (*n*, %)	0 (0)
Agree (*n*, %)	3 (11.11)
Neutral (*n*, %)	5 (18.52)
Disagree (*n*, %)	12 (44.44)
Strongly disagree (*n*, %)	7 (25.93)
My patients are satisfied with the available treatment options for MPS. (*n* = 27)	
Strongly agree (*n*, %)	0 (0)
Agree (*n*, %)	4 (14.81)
Neutral (*n*, %)	8 (29.63)
Disagree (*n*, %)	12 (44.44)
Strongly disagree (*n*, %)	3 (11.11)

**Table 3 pharmaceuticals-16-00416-t003:** Results of the sample with the knowledge and use of ITTs.

Have you Ever Heard of n-of-1 Trials? (*n* = 27)	
Yes (*n*, %)	20 (74.07)
No (*n*, %)	7 (25.93)
Have you ever used an n-of-1 trial in the treatment of your MPS patients? (*n* = 27)	
Yes (*n*, %)	10 (37.04)
No (*n*, %)	17 (62.96)
Have you conducted n-of 1 trials with patients suffering from other diseases? (*n* = 27)	
Yes (*n*, %)	10 (37.04)
No (*n*, %)	17 (62.96)
With which type of MPS do you ran the n-of-1 trial? (*n* = 8)	
MPS I Hurler (*n*, %)	2 (25.00)
MPS II (*n*, %)	3 (37.50)
MPS III B (*n*, %)	1 (12.50)
MPS VI (*n*, %)	1 (12.50)
MPS VII (*n*, %)	1 (12.50)
How many n-of-1 trials have you ever ran with MPS patients? (*n* = 9)	
1 (*n*, %)	7 (77.8)
2 (*n*, %)	1 (11.11)
3 (*n*, %)	1 (11.11)
4 (*n*, %)	0 (0)
5 (*n*, %)	0 (0)
> 5 (*n*, %)	0 (0)
Have you ran systematic evaluated n-of-1 trials with an advanced monitoring plan? (*n* = 12)	
Yes (*n*, %)	4 (33.33)
No (*n*, %)	8 (66.67)
Have you included further experts in your n-of-1 trial? (*n* = 13)	
Yes (*n*, %)	5 (38.46)
No (*n*, %)	8 (61.54)
Have you ever published your n-of-trials in a journal? (*n* = 13)	
Yes (*n*, %)	2 (15.38)
No (*n*, %)	11 (84.62)
What is the primary reason why you do not use n-of-1 trials in your practice? (*n* = 21)	
Impractical to implement (*n*, %)	5 (23.81)
Not sufficiently trained in n-of-1 trial design (*n*, %)	4 (19.05)
Too time consuming (*n*, %)	4 (19.05)
Not relevant to most patients in my practice (*n*, %)	5 (23.81)
My patients are unlikely to be interested in my practice (*n*, %)	0 (0)
Some other reason (*n*, %)	3 (14.29)

**Table 4 pharmaceuticals-16-00416-t004:** Results of DAF-supported ITTs in MPS.

Assume There Is a Service That Made it Easy for You to Offer n-of-1 Trials to Select Patients in Your Practice with MPS. How Likely Are You to Use a Service Like This to Make Data Driven Treatment Choices at Least One in the Next Year? (*n* = 26)	
Highly likely (*n*, %)	12 (46.15)
Likely (*n*, %)	11 (42.30)
Neutral (*n*, %)	1 (3.85)
Unlikely (*n*, %)	1 (3.85)
Highly unlikely (*n*, %)	1 (3.85)

## Data Availability

Data is contained within the article and supplementary file.

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
