# Peer review of "Individual Treatment Trials—Do Experts Know and Use This Option to Improve the Treatability of Mucopolysaccharidosis?"

_pharmaceuticals, 2023, doi:10.3390/ph16030416_

Round 1

Reviewer 1 Report

The survey based study by Wiesinger et al evaluates the knowledge of ITT among clinicians looking after MPS patients. The manuscript is generally well written.

Major comments:

what is the next step and what do the authors plan to do with the survey results to improve the awareness of ITT among clinicians?

the data regarding the metabolic centres should/could be included in the supplementary file. This is important to see which major MPS centres were included, and which were missed. 

28 clinicians may not be representative of the MPS centres in Europe, not to mention around the world. It would be worth correlating the data from the centres with the number of MPS patients they looks after and whether they are adult or paediatric MPS patients. It is relevant and would add more information the the lack of knowledge or lack of research in MPS field in some centres. Additional information regarding the established research in these centres, even if in other LSDs, exists would be relevant and would explain some of these results.

Also, whether clinicians are internists, paediatricians, etc or geneticists or metabolic clinicians who look after patients with rare diseases in general, would add to the results section.

The study has ethical approval, so adding these information, even if in an annonymised form, would help avoid bias in the results.

Author Response

Dear Reviewer, 

thank you for your valuable feedback,

We included your comments in our manuscript: 

what is the next step and what do the authors plan to do with the survey results to improve the awareness of ITT among clinicians? 

we addressed this point in the last part of our discussion - see line 220-227.

the data regarding the metabolic centres should/could be included in the supplementary file. This is important to see which major MPS centres were included, and which were missed. 

28 clinicians may not be representative of the MPS centres in Europe, not to mention around the world. It would be worth correlating the data from the centres with the number of MPS patients they looks after and whether they are adult or paediatric MPS patients. It is relevant and would add more information the the lack of knowledge or lack of research in MPS field in some centres. Additional information regarding the established research in these centres, even if in other LSDs, exists would be relevant and would explain some of these results.

Also, whether clinicians are internists, paediatricians, etc or geneticists or metabolic clinicians who look after patients with rare diseases in general, would add to the results section.

a supplementary file has been created with all necessary information regarding the survey participants (see suppl. info 1.0). We refer to this document several times in our manuscript. The correlation of data from participated centers and number of MPS patients can be found in line 214. 

Thank you and kind regards, 

Anna-Maria Wiesinger

Reviewer 2 Report

Dear Authors,

congratulations for the originality of the paper. Analysis is well conducted and detailed. This argument is very interesting and innovative.

The survey is well edited even if only few physicians were involved. It could be helpful after publication to share this survey with many other specialists or doing collaborative studies with MetabERN or SSIEM.

We suggest to hint to gene therapy of MPSIH and to discuss better the problems related to the possibility to prescribe antinflammatory drugs as offlabel use cause of loss of reimbursement or authorization by Ethical Committe.

Author Response

Dear Reviewer, 

thank you for your valuable feedback. 

The survey is well edited even if only few physicians were involved. It could be helpful after publication to share this survey with many other specialists or doing collaborative studies with MetabERN or SSIEM.

We suggest to hint to gene therapy of MPSIH and to discuss better the problems related to the possibility to prescribe antinflammatory drugs as offlabel use cause of loss of reimbursement or authorization by Ethical Committe.

We addressed your comment in the last part of our discussion (line 220-227). 

The new manuscript has been attached here and the suppl. information with all necessary details regarding participated clinicians has been attached in the authors reply to reviewer 1. 

Thank you and kind regards, 

Anna-Maria Wiesinger

Round 2

Reviewer 1 Report

thank you